# Construction of a Yeast Cell-Based Assay System to Analyze SNAP25-Targeting Botulinum Neurotoxins

**DOI:** 10.3390/microorganisms11051125

**Published:** 2023-04-26

**Authors:** Shilin Chen, Feng Li, Guoyu Liu, Yuqing Li, Zijie Li, Yishi Liu, Hideki Nakanishi

**Affiliations:** The Key Laboratory of Carbohydrate Chemistry and Biotechnology, Ministry of Education, School of Biotechnology, Jiangnan University, Wuxi 214126, China

**Keywords:** botulinum neurotoxins, yeast cell-based assay, SNARE, SNAP25, sporulation

## Abstract

Herein, we describe a yeast cell-based assay system to analyze SNAP25-targeting botulinum neurotoxins (BoNTs). BoNTs are protein toxins, and, upon incorporation into neuronal cells, their light chains (BoNT-LCs) target specific synaptosomal N-ethylmaleimide-sensitive attachment protein receptor (SNARE) proteins, including synaptosomal-associated protein 25 (SNAP25). BoNT-LCs are metalloproteases, and each BoNT-LC recognizes and cleaves conserved domains in SNAREs termed the SNARE domain. In the budding yeast *Saccharomyces cerevisiae*, the SNAP25 ortholog Spo20 is required for production of the spore plasma membrane; thus, defects in Spo20 cause sporulation deficiencies. We found that chimeric SNAREs in which SNARE domains in Spo20 are replaced with those of SNAP25 are functional in yeast cells. The Spo20/SNAP25 chimeras, but not Spo20, are sensitive to digestion by BoNT-LCs. We demonstrate that *spo20*∆ yeasts harboring the chimeras exhibit sporulation defects when various SNAP25-targeting BoNT-LCs are expressed. Thus, the activities of BoNT-LCs can be assessed by colorimetric measurement of sporulation efficiencies. Although BoNTs are notorious toxins, they are also used as therapeutic and cosmetic agents. Our assay system will be useful for analyzing novel BoNTs and BoNT-like genes, as well as their manipulation.

## 1. Introduction

Botulinum neurotoxins (BoNTs) are protein neurotoxins that target nerve cells and inhibit neurotransmitter release [1]. BoNTs are traditionally classified into seven serotypes, BoNT/A to/G [2]. The neurotoxins are predominantly produced by *Clostridium* species, but BoNT-like genes are also found in other organisms [3,4]. Although BoNTs are notorious toxins, they are also used as therapeutic agents [5]. Structurally, BoNTs are composed of a heavy chain (HC) and light chain (LC) interconnected via a disulfide bond [2]. The HC is subdivided into two functionally distinctive domains termed the H_C_ and H_N_; these domains are required for binding to target cells and translocation of the LC into the cytosol, respectively. In the cytosol of nerve cells, LC is released from HC by cleavage of the disulfide bond [6]. LC is a metalloprotease that can cleave the synaptic vesicle fusion machinery; this activity causes an abrogation of neurotransmitter release [1].

The synaptosomal N-ethylmaleimide-sensitive attachment protein receptor (SNARE) proteins serve as membrane fusion machinery in eukaryotic cells [7]. Various SNAREs are deployed in the intracellular vesicle transport system, and the localization of each SNARE protein is strictly regulated. When membranes fuse, SNARE proteins present in the opposing membranes form a tight complex called the trans-SNARE complex or SNAREpin [8,9,10]. SNARE proteins have one or two α-helical heptad repeats called the SNARE domains [11]. For the formation of the trans-SNARE complex, four SNARE domains provided by three or four SNARE proteins generate a tight coiled-coil interaction [12]. Formation of the trans-SNARE complex can induce membrane fusion by pulling the opposing membranes into close proximity [8,9,10,13]. SNARE proteins form fusogenic trans-SNARE complexes with cognate partners. Thus, SNARE proteins provide not only mechanical force but also specificity for membrane fusions [7].

Syntaxin 1, synaptosomal-associated protein 25 (SNAP25), and VAMP2 (also referred to as synaptobrevin 2) are SNARE proteins engaged in synaptic vesicle fusion in neuronal cells [8]. Syntaxin 1 and SNAP25 are localized to the plasma membrane [14,15], whereas VAMP2 is localized to synaptic vesicles [16]. Syntaxin 1 and VAMP2 are transmembrane proteins containing a single SNARE domain. SNAP25 is a unique SNARE because it contains two SNARE domains but no transmembrane domain [12]. Synaptic SNARE proteins are targets of BoNT-LCs; each BoNT-LC recognizes the SNARE domain of a specific SNARE and cleaves at a specific site in the domain [17]. For example, BoNT/A-LC and BoNT/B-LC cleave the SNARE domain of SNAP25 and VAMP2, respectively [18,19]. Some BoNT-LCs target multiple SNAREs; for example, BoNT/C-LC cleaves the SNARE domains of SNAP25 and syntaxin 1 [20].

SNARE proteins are functionary and structurally conserved in eukaryotes. In the budding yeast *Saccharomyces cerevisiae*, Sso1/2, Sec9, and Snc1/2 are orthologs of syntaxin 1, SNAP25, and VAMP2, respectively [21] (Sso1/2 and Snc1/2 represent that these SNAREs have two functional paralogs). Sso1/2, Sec9, and Snc1/2 are required for fusion of secretary vesicles to the plasma membrane [22]. In addition to exocytic vesicle fusion, Sso1 and Snc1/2 are involved in the formation of the prospore membrane, which is a membrane uniquely generated in sporulating yeast [23]. Sporulation is a developmental process that generates quiescent and stress-resistant haploid cells from diploid cells. In this process, secretory vesicles are used to produce prospore membranes that engulf haploid nuclei generated by meiosis [23]. The prospore membrane becomes the spore plasma membrane in mature spores. For production of the prospore membrane, the function of Sec9 is replaced by another yeast SNAP25 ortholog Spo20 [24]. *SPO20* is specifically expressed in sporulating cells, and its disruptants (*spo20*∆ cells) are viable but exhibit a sporulation deficiency.

Yeast SNAREs are generally resistant to BoNT-LCs but chimeric SNAREs in which SNARE domains in yeast SNAREs are replaced with those of human orthologs become sensitive to digestion with BoNT-LCs [25,26]. Yeast cells harboring such a chimera instead of the original exocytic SNARE exhibit a growth defect upon expression of an appropriate BoNT-LC. Thus, yeast cells harboring yeast/human chimeric SNAREs can be used as an assay system to analyze BoNT-LCs. Previous studies reported Sso1/syntaxin 1 and Snc2/VAMP2 chimeras [25,26]. In these assays, the activities of BoNT-LCs were analyzed by monitoring the yeast growth rate. In this study, we constructed Spo20/SNAP25 chimeras. Unlike the previously reported yeast cell-based BoNT-LC assays, in this assay, the effects of neurotoxins are quantified by measuring sporulation efficiency, which can be determined by colorimetric measurement. The assay system can be used to find and analyze SNAP25-targeting BoNTs.

## 2. Materials and Methods

### 2.1. Plasmids

All plasmids and primers used in this study are listed in Appendix A, respectively. Detailed methods for construction of the plasmids are described in the Appendix A. Gene cloning and other experiments were performed following the Measures for Biosafety Management of Jiangnan University Laboratory.

### 2.2. Yeast Strains, Growth Media, and Cell Culture

The *S. cerevisiae* strains used in this study are listed in Appendix A. To construct the *spo20*∆ diploid strain, a DNA fragment for the gene disruption was amplified by PCR using pFA6a-His3MX6 [27] as a template and HXO26 and HXO27 as primers. The PCR fragment was integrated into AN117-4B and AN117-16D [28], and the resulting haploid strains were crossed to generate diploid cells. To construct the *sec9*∆ mutant, pRS316TEF-Sec9 was first transformed into YPH499 [29], and then genomic *SEC9* was disrupted in the transformants. For the deletion of *SEC9*, the PCR cassette for the gene deletion was generated by PCR using CSL1 and CSL2 as primers, and pFA6a-His3MX6 as a template.

To induce sporulation, yeast cells were cultured in 5 mL of YPAD liquid medium (yeast extract 10 g/L, peptone 20 g/L, adenine 30 mg/L, and glucose 20 g/L) overnight. Then, 0.1 mL of the culture was transferred into 5 mL of YPAcetate liquid medium (yeast extract 10 g/L, peptone 20 g/L, and potassium acetate 20 g/L) and cultured overnight. Cells were then centrifugated and resuspended in sporulation medium (2% potassium acetate) at a concentration of 3 × 10^7^ cells/mL and cultured for 24 h, 48 h, or 72 h at 30 °C.

### 2.3. Growth Assay

To analyze cell growth on plates, yeast cells were transformed with appropriate plasmids and the colonies were streaked on SD (6.7 g/L yeast nitrogen base, 2 g/L dropout mix without appropriate selectable supplements, and 20 g/L glucose) plates. After 3 day incubation at 30 °C, the cells were streaked on new SD plates and grown for 1 day at 30 °C. For complementation assays of Sec9/SNAP25 chimeras, *sec9*∆ cells harboring pRS316TEF-Sec9 were transformed with appropriate plasmids, and the transformants were grown on SD plates for 3 days. Then, the cells were streaked on SD plates supplemented with 1 mg/mL 5-fluoroortic acid (5-FOA) and grown for 2 days at 30 °C.

To draw growth curves, yeast cells were precultured in SD medium overnight to an OD_660_ of approximately 1.0. Cells were then centrifugated and resuspended in 20 mL of SD medium to an OD_660_ of approximately 0.1. OD_660_ was measured every 8 h.

### 2.4. Sporulation Assay

To count spores, cells in sporulation medium (0.1 mL) were collected by centrifugation, washed with water once, and resuspended in 100 μL of water. Under the light microscope, the number of asci were counted among cells found in certain area (field of view). This analysis was performed in several areas, and at least 500 cells were counted. Asci and other cells were discriminated by their morphology; asci were recognized as cells containing 2–4 smooth and spherical spores.

For the colorimetric assay, spores were generated in sporulation medium for 48 h as described above. Then, 0.2 mL of the spore suspension was taken into a 96-well plate, and fluorescence quantification was performed with a microplate reader (Synergy H4, BioTek, Winooski, VT, USA) at an excitation wavelength of 285 nm and an emission wavelength of 425 nm. A 2% potassium acetate solution was used as a blank control. Measured values were directly used as indices to represent sporulation efficiencies.

### 2.5. Western Blotting

Cell extracts were made by disruption with glass beads in urea buffer (8 M urea and 1 mM phenylmethylsulphonyl fluoride) followed by 10 min of centrifugation at 21,500× *g*. Protein concentrations in the supernatants were determined using a Nano-Drop (Thermo-Scientific, Shanghai, China). Then, 100 μg of proteins were subjected to 10% SDS-PAGE and transferred to PVDF membranes (Bio-Rad, Shanghai, China). Mouse anti-FLAG (Transgen, Beijing, China), mouse anti-GFP (Transgen, Beijing, China), mouse anti-HA (Transgen, Beijing, China), or mouse anti-actin (Transgen, Beijing, China) antibodies were used as primary antibodies at 1:5000 dilutions. Goat anti-mouse IgG-HRP (Transgen, Beijing, China) was used as a secondary antibody at 1:5000 dilution. Signals were visualized by Clarity Western ECL Substrate (Bio-Rad, Shanghai, China). Tanon-5200Multi (Tanon, Shanghai, China) was used to obtain Western blot images. Quantification of the signals were performed with ImageJ software [30]. Intensities of target protein band were normalized to actin (the intensity of the target protein band was divided by the relative intensity of the corresponding actin band). Then, relative intensities of target protein bands were calculated.

### 2.6. Microscopy

Microscopy images were obtained using a Nikon C2 Eclipse Ti-E (Nikon, Shanghai, China) inverted microscope with a DS-Ri camera equipped with NIS-Element AR software.

### 2.7. Statistics

All experiments were performed with three or four independent samples; cells from different colonies were cultured on different days. Differences between the analyzed samples were considered significant at *p* < 0.05. Statistical significance was determined using a two-tailed unpaired Student’s *t*-test calculated with Prism 9.0.0 software.

## 3. Results

### 3.1. Production of Functional Chimeras of Yeast and Human SNAP25

Deletion of *SEC9* causes a lethal phenotype in yeast cells, which was not complemented by expression of SNAP25 (Appendix A). To establish a yeast system to analyze BoNT-LCs targeting SNAP25, we constructed Sec9/SNAP25 chimeras. Sec9 has two SNARE domains; the N- and C-terminal SNARE domains are referred to as SNARE(N) and SNARE(C), respectively (Appendix A). To this end, we first constructed a chimera in which SNARE(C) in Sec9 was replaced with that of SNAP25 (Appendix A). To detect the chimera, green fluorescent protein (GFP) was added to the N-terminus (the fusion protein is referred to as GFP-Sec9/SNAP25(C)). The *sec9*∆ cells harboring GFP-Sec9 grew, showing that the addition of GFP at the N-terminus of Sec9 did not abrogate the function of Sec9 (Appendix A). While the levels of GFP-Sec9/SNAP25(C) expressed in *sec9*∆ cells were similar to those of GFP-Sec9 (Appendix A), the cells harboring GFP-Sec9/SNAP25(C) failed to grow (Appendix A). Sec9/SNAP25(C) chimera without any tagging did not complement *sec9* deletion either (Appendix A). To assess which amino-acid residues in the SNARE domain are required for Sec9 function in the SNARE(C) region, we constructed a series of chimeras in which various parts of the SNARE(C) region were replaced with the corresponding regions of SNAP25 (schematic diagram is shown in Appendix A). The results of this analysis showed that only seven amino acids of SNARE(C) in Sec9 were exchanged to produce a functional chimera (Appendix A). Thus, the function of the SNARE domain in Sec9 was hardly substituted by that of SNAP25.

*S. cerevisiae* has another SNAP25 ortholog Spo20 [24]. Thus, we examined whether the SNARE domains of SNAP25 were functional in Spo20. We constructed a Spo20/SNAP25 chimera in which SNARE(C) was replaced with the human ortholog (Figure 1a). Instead of GFP, a hemagglutinin (HA) tag was added to the N-terminus of the chimera since the small epitope rarely compromises protein functions. The fusion protein was termed HA-Spo20/SNAP25(C). HA-Spo20/SNAP25(C) was placed under the control of the sporulation-specific *SPO20* promoter, and the expression plasmid was integrated into the genome of *spo20*∆ cells. While *spo20*∆ cells exhibited sporulation deficiency, the defect was recovered by expression of HA-Spo20/SNAP25(C) (Figure 1b and Appendix A). Compared to the *spo20*∆ mutant harboring HA-Spo20, those harboring HA-Spo20/SNAP25(C) took a longer time until the sporulation rate reached a plateau (Figure 1b). Nevertheless, *spo20*∆ mutant cells harboring HA-Spo20 and HA-Spo20/SNAP25(C) formed spores at comparable levels when the cells were incubated in sporulation media for 48 h (Figure 1b). Thus, in this study, sporulation efficiencies were measured after 48 h of incubation in sporulation medium.

Next, we examined a Spo20/SNAP25 chimera in which both SNARE domains in Spo20 were replaced with those of SNAP25 (the chimera was named Spo20/SNAP25(NC)) (Figure 1a). However, the HA-Spo20/SNAP25(NC) chimera was not functional in *spo20*∆ mutants (Figure 1c). We constructed another chimera in which the SNARE(N) domain in Spo20 was replaced with that of SNAP25 (the chimera was named Spo20/SNAP25(N)) (Figure 1a). HA-Spo20/SNAP25(N) rescued the sporulation defect in *spo20*∆ mutants (Figure 1c). Although the sporulation efficiency of *spo20*∆ mutant cells harboring HA-Spo20/SNAP25(N) was decreased by 14% compared to those harboring HA-Spo20, more than 80% of the cells harboring the chimera formed spores (Figure 1c).

### 3.2. The Spo20/SNAP25(C) Chimera Is Targeted by BoNT/C-LC

To examine whether Spo20/SNAP25 chimeras were targeted by BoNT-LCs, BoNT/C-LC was expressed under the control of the constitutive *TEF1* promoter in *spo20*∆ cells harboring HA-Spo20/SNAP25(C) (Figure 2a). In HA-Spo20-harboring cells, sporulation efficiency was not altered by the expression of BoNT/C-LC (Figure 2b). However, in HA-Spo20/SNAP25(C)-harboring cells, sporulation efficiencies were decreased by 40% by expression of BoNT/C-LC (Figure 2a). To verify that the decrease in sporulation efficiency was attributable to the proteolytic activity of BoNT/C-LC, a catalytically inactive mutant, BoNT/C^E230Q^-LC [31], was expressed in HA-Spo20/SNAP25(C)-harboring cells (Figure 2c). As shown in Figure 2c, sporulation efficiency was not decreased by expression of the catalytically inactive BoNT/C-LC mutant. It should be noted that expression levels of BoNT/C^E230Q^-LC were decreased by 83% compared to those of wildtype (Appendix A). To further verify that HA-Spo20/SNAP25(C) was cleaved by BoNT/C-LC, we performed Western blotting to detect the chimera. However, HA-Spo20/SNAP25(C) expressed from the plasmid integrated into the genome was not detected (Appendix A). Thus, Spo20/SNAP25(C) fused to a 3× HA tag at the C-terminus (Spo20/SNAP25(C)-3HA) was overexpressed from a multicopy vector in *spo20*∆ cells. As shown in Appendix A, Spo20/SNAP25(C)-3HA was detected in sporulating cells. The expression levels of Spo20/SNAP25(C)-3HA in *spo20*∆ cells reached the highest value after 12 h of incubation in sporulation medium (Appendix A). In BoNT/C-LC-expressing cells but not in BoNT/C^E230Q^-LC-expressing cells, the protein levels of Spo20/SNAP25(C)-3HA were decreased, presumably due to digestion by the neurotoxin (Figure 2d). The levels of Spo20-3HA were not altered by the expression of BoNT/C-LC (Figure 2e).

### 3.3. Yeast Cells Harboring the Spo20/SNAP25 Chimera Can Be Used to Assay BoNT-LCs

Spores contain a fluorescent molecule, bisformyldityrosine, in the spore wall [32]. We found that the quantities of spores were correlated with the fluorescent levels of dityrosine (Appendix A). While spores were counted under a microscope in the experiments described above, the obtained result suggests that sporulation efficiency can be quantified more readily and unbiasedly by the colorimetric measurement. To test this possibility, the sporulation efficiencies of the cells described in Figure 2a,c were measured by colorimetric assay. Appendix A shows that the microscopy and colorimetric assays exhibited similar results (compare Figure 2a,c and Appendix A). Thus, hereafter, the colorimetric assay was employed to measure sporulation efficiencies.

To further examine whether Spo20/SNAP25 chimeras can be used to analyze BoNT-LCs, BoNT/A-LC, which is another SNAP25 targeting BoNT-LC, was expressed in *spo20*∆ cells harboring HA-Spo20/SNAP25(C) from the *TEF1* promoter (Figure 3a). The expression of BoNT/A-LC caused a 65% decrease in sporulation efficiency in the HA-Spo20/SNAP25(C)-harboring cells (Figure 3a). A severe sporulation defect was not observed when a catalytically inactive BoNT/A-LC mutant, BoNT/A^E224Q^-LC, was expressed in the HA-Spo20/SNAP25(C)-harboring cells, although its expression caused a slight decrease (by 15%) in sporulation efficiency (Figure 3a). Notably, expression levels of the wildtype and mutant BoNT/A-LC were comparable (Figure 3a and Appendix A). Levels of Spo20/SNAP25(C)-3HA were decreased by expression of BoNT/A-LC but not by BoNT/A^E224Q^-LC (Figure 3b). In *spo20*∆ cells harboring HA-Spo20, sporulation efficiencies were not altered by the expression of BoNT/A-LC (Figure 3c). The expression of BoNT/A-LC did not affect the levels of Spo20-3HA detected in *spo20*∆ cells (Figure 3d).

BoNT/B-LC targets VAMP2 but not SNAP25. Accordingly, expression of BoNT/B-LC did not inhibit sporulation of HA-Spo20/SNAP25(C)-harboring *spo20*∆ cells (Appendix A). Spo20/SNAP25(C)-3HA levels were not altered by expression of BoNT/B-LC (Appendix A).

### 3.4. Use of Yeast Cells Harboring Spo20/SNAP25 Chimeras to Analyze BoNT/E Family Members

BoNT/E1 is another botulinum toxin that cleaves SNARE(C) in SNAP25 [18]. When BoNT/E1-LC was expressed in wildtype cells from the *TEF1* promoter, we found that growth of the yeast cells became slower compared to those harboring the empty vector (Appendix A). Thus, BoNT/E1-LC was expressed from the *SPO20* promoter so that the expression of BoNT-LC was restricted in sporulating cells (Figure 4a). The *spo20*∆ cells harboring *SPO20* promoter-driven BoNT/E1-LC grew normally (Appendix A). Importantly, sporulation of *spo20*∆ cells harboring HA-Spo20 cells was not inhibited by expression of BoNT/E1-LC (Appendix A). However, in *spo20*∆ cells harboring HA-Spo20/SNAP25(C), sporulation efficiency was decreased by 67% by expression of BoNT/E1-LC (Figure 4b). Western blotting results showed that the levels of Spo20/SNAP25(C)-3HA were decreased by the expression of BoNT/E1-LC (Figure 4c). By expression of a catalytically inactive BoNT/E1-LC mutant, BoNT/E1^E213Q^-LC, sporulation efficiency and levels of the chimeric SNARE were not altered (Figure 4c,d), although expression levels of BoNT/E1^E213Q^-LC were decreased by 46% compared to those of wildtype (Appendix A). BoNT/E1-LC does not cleave SNARE(N) in SNAP25. Consistent with this specificity, sporulation of HA-Spo20/SNAP25(N)-harboring *spo20*∆ cells was not inhibited by expression of BoNT/E1-LC (Figure 4e).

BoNT/E12 is a BoNT/E subtype but its catalytic activity has not been previously analyzed [33]. Thus, BoNT/E12-LC was expressed in *spo20*∆ cells harboring HA-Spo20/SNAP25(C) (Figure 4b). Since expression of BoNT/E12-LC from the *TEF1* promoter also caused a growth defect (Appendix A), the protein was expressed from the *SPO20* promoter. In our assay, BoNT/E12-LC exhibited similar activity to BoNT/E1-LC; a sporulation defect was observed in *spo20*∆ cells harboring HA-Spo20/SNAP25(C), but not in those harboring HA-Spo20 or HA-Spo20/SNAP25(N), by expression of BoNT/E12-LC (Figure 4b,e and Appendix A). Spo20/SNAP25(C)-3HA levels were decreased by the expression of BoNT/E12-LC (Figure 4c). A catalytically inactive mutant, BoNT/E12^E213Q^-LC, did not inhibit sporulation of the HA-Spo20/SNAP25(C)-harboring cells (Figure 4d). Levels of Spo20/SNAP25(C)-3HA were not decreased by expression of BoNT/E12^E213Q^-LC (Figure 4c). Expression levels of BoNT/E12^E213Q^-LC were decreased by 19% compared to wildtype (Appendix A).

### 3.5. Analysis of BoNT/En-LC, Which Cleaves SNARE(N) in SNAP25

BoNT/En is a BoNT-like protein found in *Enterococcus* spp. [4]. BoNT/En-LC is known to cleave SNARE(N) in SNAP25. Thus, to examine whether HA-Spo20/SNAP25(N) could be targeted by BoNT-LCs, we expressed BoNT/En-LC from the *TEF1* promoter in wildtype cells. However, like BoNT/E1-LC and/E12-LC, the growth of the yeast cells also became slower than that of those harboring the empty vector (Appendix A). Thus, BoNT/En-LC was expressed from the *SPO20* promoter, which did not interfere with the vegetative growth (Appendix A). In *spo20*∆ cells harboring HA-Spo20, sporulation was not compromised by the *SPO20* promoter-driven BoNT/En-LC (Appendix A). However, in *spo20*∆ cells harboring HA-Spo20/SNAP25(N), sporulation efficiency was decreased by expression of BoNT/En-LC by 81% (Figure 5a). The sporulation efficiency of the HA-Spo20/SNAP25(C)-harboring cells was not decreased by BoNT/En-LC expression (Figure 5b). These results are consistent with the notion that BoNT/En-LC targets SNAP25(N) but not SNAP25(C). Additionally, the levels of Spo20/SNAP25(N)-3HA, but not Spo20/SNAP25(C)-3HA, were decreased by the expression of BoNT/En-LC (Figure 5c,d). We constructed a putative catalytically inactive BoNT/En-LC mutant by mutating the conserved Gln226 residue to Gln. Expression of BoNT/En^E226Q^-LC caused a slight decrease in sporulation efficiency (by 10.4%) in the HA-Spo20/SNAP25(C)-harboring cells; however, the effect was relatively mild compared to that of BoNT/En-LC expression (Figure 5a). Protein levels of Spo20/SNAP25(N) were not altered by BoNT/En^E226Q^-LC expression (Figure 5c). Expression levels of BoNT/En^E226Q^-LC were decreased by 68% compared to those of wildtype (Appendix A).

## 4. Discussion

We found that SNARE domains in Spo20 are amenable to exchange with those of SNAP25, which allowed us to construct a yeast cell-based assay system to analyze SNAP25 targeting BoNT-LCs. Previously, we and other groups reported yeast cell-based BoNT-LC assays in which the activities of neurotoxins were analyzed by monitoring the yeast growth. In this study, we devised an assay to use sporulation efficiency as an indicator of the activity of BoNT-LCs. One advantage of this assay is that SNAP25-targeting BoNT-LCs are readily identified.

Sec9 is another yeast SNAP25 ortholog, and Spo20 and Sec9 share partner SNAREs [23]. However, we found that Sec9 becomes nonfunctional when SNARE(C) is replaced with that of SNAP25. Given that Spo20/SNAP25 chimeras are functional in yeast cells, SNARE domains of SNAP25 can form trans-SNARE complexes with partner SNAREs. A previous study showed that Sec9 and Spo20 have distinct properties; Sec9 produces a tight trans-SNARE complex compared to Spo20 [34]. These SNAREs have distinct binding affinity because vesicle fusion at the plasma membrane requires more energy to generate the prospore membrane [34]. Thus, the SNARE domain in Sec9 is unable to be replaced with that of SNAP25, presumably because the SNARE complex containing a SNAP25 SNARE domain is not sufficient to induce vesicle fusion at the plasma membrane. Regulation of synaptic SNARE proteins is more complex compared to that of yeast exocytic SNARE proteins. The previous study using Sso1/syntaxin 1A revealed that syntaxin-1A and Sso1 exhibit distinctive properties [35]. Thus, further analysis of yeast/human chimeric SNAREs, including Spo20/SNAP25, would provide further insight into the regulatory mechanism of synaptic SNARE proteins.

We examined various BoNT-LCs in our assay system; generally, the results are consistent with the previously reported specificities of BoNT-LCs. Of note, the expression of catalytically inactive BoNT/A^E224Q^-LC and/En^E226Q^-LC caused a slight decrease in *spo20*∆ cells harboring Spo20/SNAP25 chimeras. Levels of Spo20/SNAP25 chimeras were not altered by their expression. While the reason why the mutant BoNT-LCs exhibit the inhibitory effect is not clear, a similar phenomenon was reported in vivo; mice treated with a high dose of a catalytically inactive BoNT/C exhibited neuromuscular impairment [36]. Thus, BoNT-LCs may be able to compromise SNARE function even without cleaving them; for example, their binding to the target may inhibit SNARE assembly. Through mutational analysis, we showed that the catalytic activity is required to exhibit toxicity in yeast cells for BoNT/A-LC. However, for other BoNT-LCs, expression levels of catalytically inactive mutants were lower compared to those of wildtype ones. Nevertheless, our results overall showed that the BoNT-LCs target appropriate chimeras in yeast cells.

Expression of BoNT/En-LC from a constitutive promoter causes growth defects in yeast cells. Since BoNT/En-LC is known to target multiple SNAREs, including SNAP25 and VAMP2 [4], yeast SNARE(s) may be targeted by the BoNT-LC, which is a probable reason for the growth defect. BoNT/E12-LC is highly identical to BoNT/E1-LC, and our results showed that their LCs have a similar substrate specificity to that of BoNT/E1-LC. Like BoNT/En-LC, the expression of BoNT/E1 and/E12-LCs induces growth defects in yeast cells. Thus, while BoNT/E1-LC is known to cleave only SNAP25 in mammalian cells [37], BoNT/E family members may target SNAREs in other organisms. Notably, when these BoNT-LCs were expressed from the *SPO20* promoter, cells harboring wildtype Spo20 showed neither growth nor sporulation defects. Thus, the expression of BoNT-LCs from *SPO20* or other sporulation-specific promoters is likely a good option to improve the assay system.

Sso1/Syntain 1A and Snc2/VAMP2 chimeras were previously produced [25,26]. However, in the Snc2/VAMP2 chimera, only a part of the VAMP2 SNARE domain was introduced in the yeast ortholog [25]. Thus, if a functional Snc2/VAMP2 chimera containing the entire VAMP2 SNARE domain is constructed, a yeast cell-based comprehensive BoNT-LC assay system could be established. Recent advances in genome mining have revealed novel BoNTs and BoNT-like genes. The yeast cell-based assay system will be useful for characterizing novel neurotoxins. Furthermore, the assay system could be used to screen BoNT-LCs with modified activities.

## Figures and Tables

**Figure 1 microorganisms-11-01125-f001:**
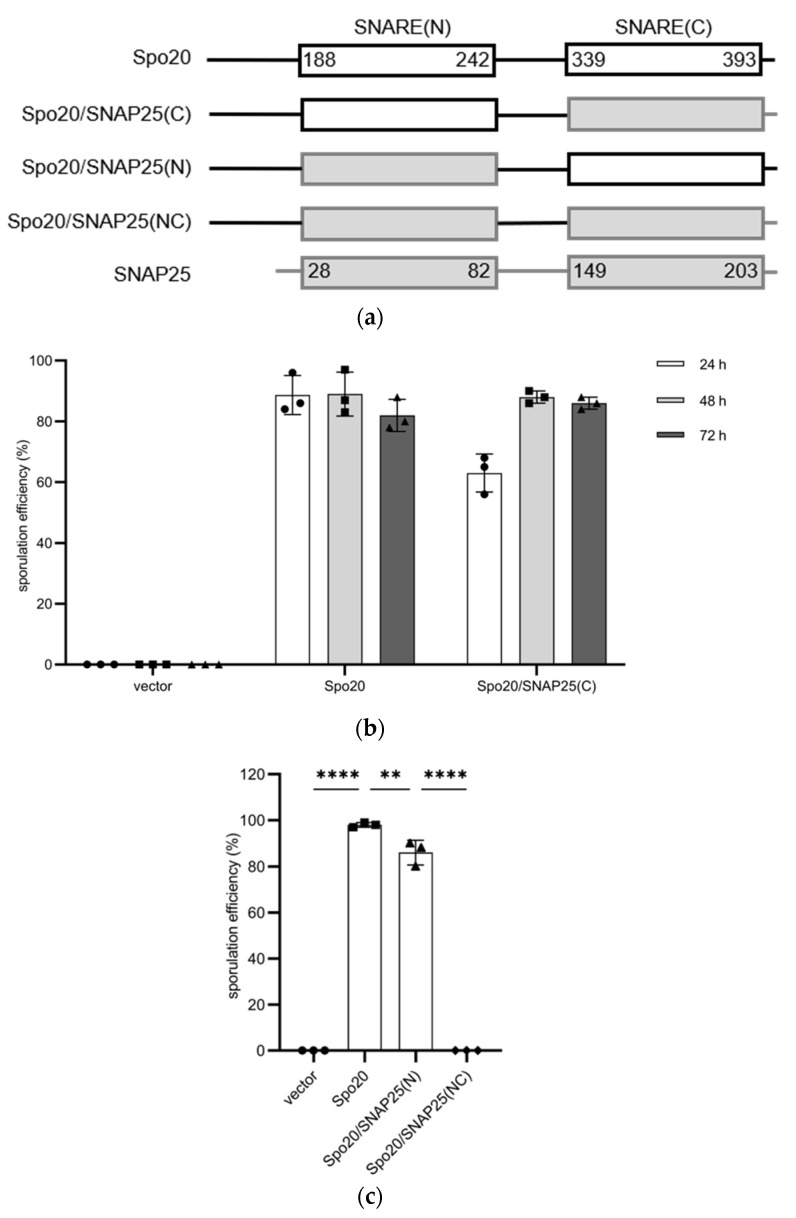
Spo20/SNAP25 chimeras are functional in yeast cells. (**a**) Schematic representations of Spo20, SNAP25, Spo20/SNAP25(C), Spo20/SNAP25(N), and Spo20/SNAP25(NC). Numbers indicate amino-acid positions. White and gray boxes represent SNARE domains derived from Spo20 or SNAP25, respectively. (**b**) *spo20*∆ cells transformed with pRS306-SPO20pr (vector), pRS306-SPO20pr-HA-Spo20 (Spo20), or pRS306-SPO20pr-HA-Spo20/SNAP25(C) (Spo20/SNAP25(C)) were cultured in sporulation medium for the indicated times, and spores were counted under the microscope. (**c**) *spo20*∆ cells transformed with pRS306-SPO20pr (vector), pRS306-SPO20pr-HA-Spo20 (Spo20), pRS306-SPO20pr-HA-Spo20/SNAP25(N) (Spo20/SNAP25(N)), or pRS306-SPO20pr-HA-Spo20/SNAP25(NC) (Spo20/SNAP25(NC)) were cultured in sporulation medium for 48 h, and spores were counted under the microscope. Data are presented as the mean ± SEM (**b**,**c**). Statistical significance was determined by two-tailed unpaired Student’s *t*-tests; *n* = 3 (**b**,**c**); ** *p* < 0.01, **** *p* < 0.0001.

**Figure 2 microorganisms-11-01125-f002:**
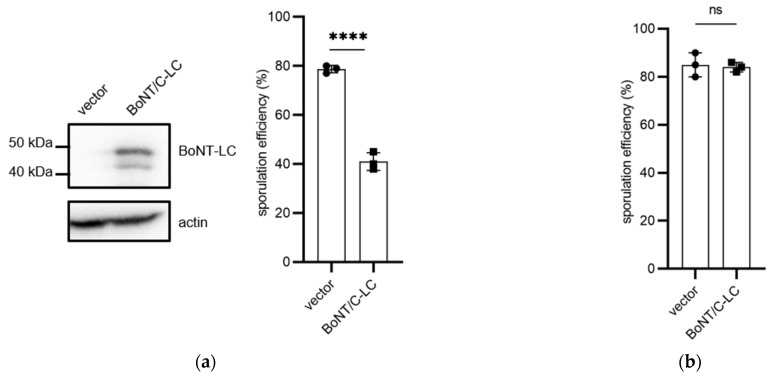
Expression of BoNT/C-LC in yeast cells harboring Spo20/SNAP25(C) chimera. (**a**) The *spo20*∆ cells harboring HA-Spo20/SNAP25(C) were transformed with pRS424-TEFpr (vector) or pRS424-TEFpr-BoNT/C-LC-FLAG (BoNT/C-LC). Left panel: these cells were lysed, and the lysates were subjected to Western blot analysis using anti-FLAG (BoNT-LC) and anti-actin antibodies. Right panel: the cells were incubated in sporulation medium for 48 h, and spores were counted under the microscope. (**b**) The *spo20*∆ cells harboring HA-Spo20 were transformed with pRS424-TEFpr (vector) or pRS424-TEFpr-BoNT/C-LC-FLAG (BoNT/C-LC); the cells were incubated in sporulation medium for 48 h, and spores were counted under the microscope. (**c**) The *spo20*∆ cells harboring HA-Spo20/SNAP25(C) were transformed with pRS424-TEFpr (vector) or pRS424-TEFpr-BoNT/C^E230Q^-LC-FLAG (mutBoNT/C-LC). Left panel: the cells were lysed and subjected to Western blot analysis using anti-FLAG (BoNT-LC) and anti-actin antibodies. Right panel: the cells were incubated in sporulation medium for 48 h, and spores were counted under the microscope. (**d**) The *spo20*∆ cells harboring Spo20/SNAP25(C)-3HA were transformed with pRS424-TEFpr (vector), pRS424-TEFpr-BoNT/C-LC-FLAG (wt), or pRS424-TEFpr-BoNT/C^E230Q^-LC-FLAG (mut). Left panel: the cell lysates were analyzed by Western blot analysis using anti-HA (Spo20/SNAP25(C)) and anti-actin antibodies. Right panel: the relative intensities of Spo20/SNAP25(C)-3HA detected in the cell lysates are shown. Intensities of the Spo20/SNAP25(C)-3HA band were normalized to actin. Their relative intensities were calculated by reference to the band in pRS424-TEFpr-harboring cells. (**e**) The *spo20*∆ cells harboring Spo20-3HA were transformed with pRS424-TEFpr (vector) or pRS424-TEFpr-BoNT/C-LC-FLAG (BoNT/C-LC). Left panel: the cell lysates were analyzed by Western blot analysis using anti-HA (Spo20) and anti-actin antibodies. Right panel: the relative intensities of Spo20-3HA detected in the cell lysates are shown. Intensities of the Spo20-3HA band were normalized to actin. The intensity of the Spo20-3HA band detected in the cells harboring pRS424-TEFpr was defined as 1. Data are presented as the mean ± SEM. Statistical significance was determined by two-tailed unpaired Student’s *t*-tests; *n* = 3 (**a**–**c**), *n* = 4 (**d**,**e**); **** *p* < 0.0001, ns: not significant (*p* ≥ 0.05).

**Figure 3 microorganisms-11-01125-f003:**
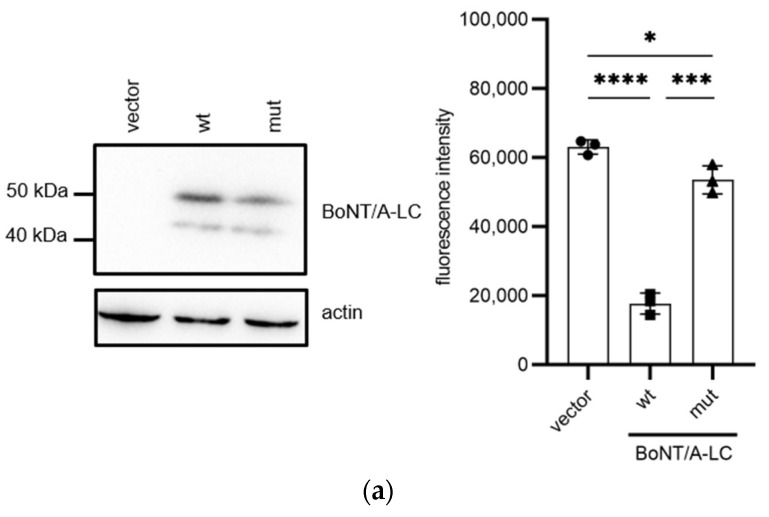
Expression of BoNT/A-LC in yeast cells harboring Spo20/SNAP25(C) chimera. (**a**) The *spo20*∆ cells harboring HA-Spo20/SNAP25(C) were transformed with pRS424-TEFpr (vector), pRS424-TEFpr-BoNT/A-LC-FLAG (wt), or pRS424-TEFpr-BoNT/A^E224Q^-LC-FLAG (mut). Left panel: the cell lysates were subjected to Western blot analysis using anti-FLAG (BoNT/A-LC) and anti-actin antibodies. Right panel: the cells were incubated in sporulation medium for 48 h, and sporulation efficiencies were measured with the colorimetric assay. (**b**) The *spo20*∆ cells harboring Spo20/SNAP25(C)-3HA were transformed with pRS424-TEFpr (vector), pRS424-TEFpr-BoNT/A-LC-FLAG (wt), or pRS424-TEFpr-BoNT/A^E224Q^-LC-FLAG (mut). Left panel: cell lysates were subjected to Western blot analysis using anti-HA (Spo20/SNAP25(C)) and anti-actin antibodies. Right panel: the relative intensities of Spo20/SNAP25(C)-3HA detected in the cell lysates are shown. The intensity of the Spo20/SNAP25(C)-3HA band was normalized to actin. Their relative intensities were calculated by reference to the band in pRS424-TEFpr-harboring cells. (**c**) The *spo20*∆ cells harboring HA-Spo20 were transformed with pRS424-TEFpr (vector) or pRS424-TEFpr-BoNT/A-LC-FLAG (BoNT/A-LC), and their sporulation efficiencies were measured with the colorimetric assay. (**d**) The *spo20*∆ cells harboring Spo20-3HA were transformed with pRS424-TEFpr (vector) or pRS424-TEFpr-BoNT/A-LC-FLAG (BoNT/A-LC). Left panel: cell lysates were subjected to Western blot analysis using anti-HA (Spo20) and anti-actin antibodies. Right panel: the relative intensities of Spo20-3HA detected in the cell lysates are shown. The intensities of the Spo20-3HA band were normalized to actin. The intensity of the Spo20-3HA band detected in the cells harboring pRS424-TEFpr was defined as 1. Data are presented as the mean ± SEM. Statistical significance was determined by two-tailed unpaired Student’s *t*-tests; *n* = 3 (**a**,**c**), *n* = 4 (**b**,**d**); * *p* < 0.05, *** *p* < 0.001, **** *p* < 0.0001, ns: not significant (*p* ≥ 0.05).

**Figure 4 microorganisms-11-01125-f004:**
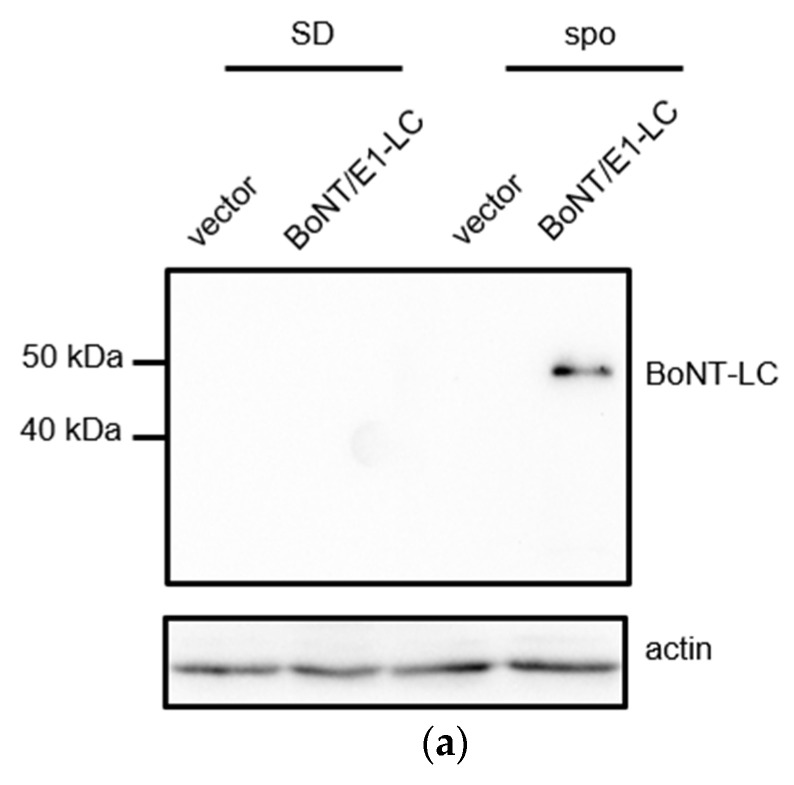
Expression of BoNT/E1-LC and/E12-LC in yeast cells harboring Spo20/SNAP25 chimeras. (**a**) The *spo20*∆ cells harboring HA-Spo20/SNAP25(C) were transformed with pRS424-SPO20pr (vector), or pRS424-SPO20pr-BoNT/E1-LC-FLAG (BoNT/E1-LC). The cells were incubated in SD or sporulation (spo) medium for 12 h. Cell lysates were subjected to Western blot analysis using anti-FLAG (BoNT/E1-LC) and anti-actin antibodies. (**b**) The *spo20*∆ cells harboring HA-Spo20/SNAP25(C) were transformed with pRS424-SPO20pr (vector), pRS424-SPO20pr-BoNT/E1-LC-FLAG (BoNT/E1-LC), or pRS424-SPO20pr-BoNT/E12-LC-FLAG (BoNT/E12-LC). Left panel: the cell lysates were subjected to Western blot analysis using anti-FLAG (BoNT-LC) and anti-actin antibodies. Right panel: the cells were incubated in sporulation medium for 48 h, and sporulation efficiencies were measured with the colorimetric assay. (**c**) The *spo20*∆ cells harboring Spo20/SNAP25(C)-3HA were transformed with pRS424-SPO20pr (vector), pRS424-SPO20pr-BoNT/E1-LC-FLAG (wt, BoNT/E1-LC), pRS424-SPO20pr-BoNT/E1^E213Q^-LC-FLAG (mut, BoNT/E1-LC), pRS424-SPO20pr-BoNT/E12-LC-FLAG (wt, BoNT/E12-LC), or pRS424-SPO20pr-BoNT/E12^E213Q^-LC-FLAG (mut, BoNT/E12-LC). Left panel: the cell lysates were subjected to Western blot analysis using anti-HA (Spo20/SNAP25(C)) and anti-actin antibodies. Right panel: relative intensities of Spo20/SNAP25(C)-3HA detected in the cell lysates. Intensities of the Spo20/SNAP25(C)-3HA band were normalized to actin. Their relative intensities were calculated by reference to the band in pRS424-SPO20pr-harboring cells. (**d**) The *spo20*∆ cells harboring HA-Spo20/SNAP25(C) were transformed with pRS424-SPO20pr (vector), pRS424-SPO20pr-BoNT/E1^E213Q^-LC-FLAG (mutBoNT/E1-LC), or pRS424-SPO20pr-BoNT/E12^E213Q^-LC-FLAG (mutBoNT/E12-LC). Left panel: the cell lysates were subjected to Western blot analysis using anti-FLAG (BoNT-LC) and anti-actin antibodies. Right panel: the cells were incubated in sporulation medium for 48 h, and sporulation efficiencies were measured with the colorimetric assay. (**e**) The *spo20*∆ cells harboring HA-Spo20/SNAP25(N) were transformed with pRS424-SPO20pr (vector), pRS424-SPO20pr-BoNT/E1-LC-FLAG (BoNT/E1-LC), or pRS424-SPO20pr-BoNT/E12-LC-FLAG (BoNT/E12-LC). The cells were incubated in sporulation medium for 48 h, and sporulation efficiencies were measured with the colorimetric assay. Data are presented as the mean ± SEM (**b**–**e**). Statistical significance was determined by two-tailed unpaired Student’s *t*-tests (**b**–**e**); *n* = 3 (**b**,**d**,**e**), *n* = 4 (**c**); ** *p* < 0.01, **** *p* < 0.0001, ns: not significant (*p* ≥ 0.05).

**Figure 5 microorganisms-11-01125-f005:**
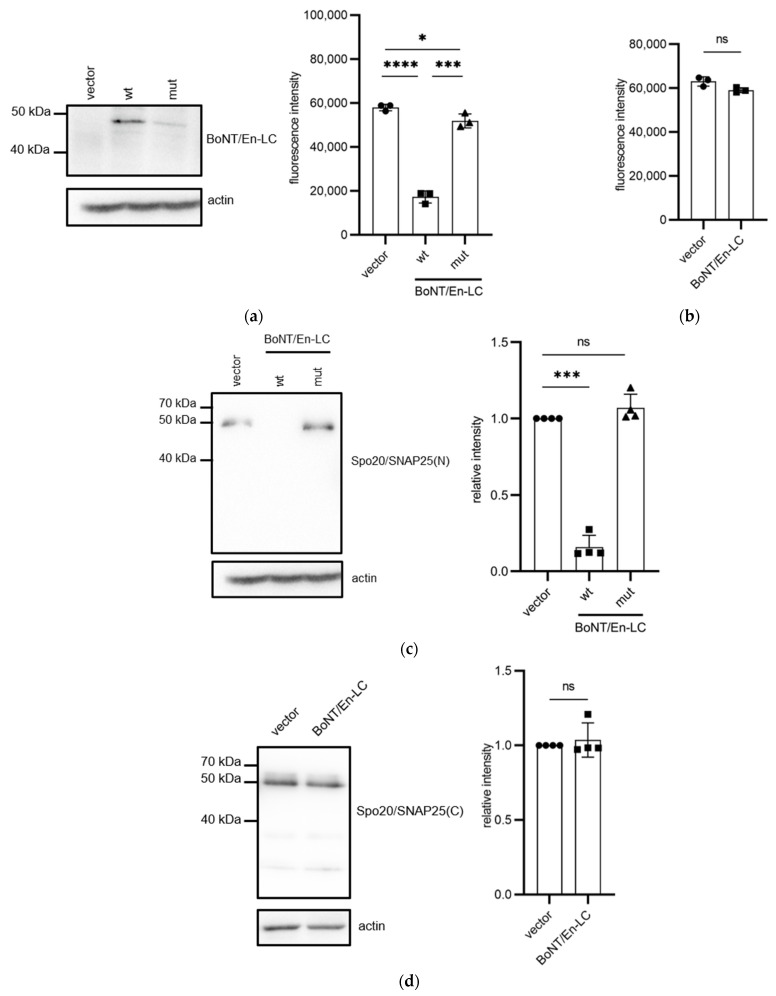
Expression of BoNT/En-LC in yeast cells harboring Spo20/SNAP25 chimeras. (**a**) The *spo20*∆ cells harboring HA-Spo20/SNAP25(N) were transformed with pRS424-SPO20pr (vector), pRS424-SPO20pr-BoNT/En-LC-FLAG (wt), or pRS424-SPO20pr-BoNT/En^E226Q^-LC-FLAG (mut). Left panel: the cell lysates were subjected to Western blot analysis using an anti-FLAG (BoNT/En-LC) and anti-actin antibodies. Right panel: The cells were incubated in sporulation medium for 48 h, and sporulation efficiencies were measured with the colorimetric assay. (**b**) The *spo20*∆ cells harboring HA-Spo20/SNAP25(C) were transformed with pRS424-SPO20pr (vector) or pRS424-SPO20pr-BoNT/En-LC-FLAG (BoNT/En-LC). The cells were incubated in sporulation medium for 48 h, and sporulation efficiencies were measured with the colorimetric assay. (**c**) The *spo20*∆ cells harboring Spo20/SNAP25(N)-3HA were transformed with pRS424-SPO20pr (vector), pRS424-SPO20pr-BoNT/En-LC-FLAG (wt), or pRS424-SPO20pr-BoNT/En^E226Q^-LC-FLAG (mut). Left panel: the cell lysates were subjected to Western blot analysis using anti-HA (Spo20/SNAP25(N)) and anti-actin antibodies. Right panel: the relative intensities of Spo20/SNAP25(N)-3HA detected in the cell lysates are shown. The intensities of the Spo20/SNAP25(N)-3HA band were normalized to actin. Their relative intensities were calculated by reference to the band in pRS424-SPO20pr-harboring cells. (**d**) *spo20*∆ cells harboring Spo20/SNAP25(C)-3HA were transformed with pRS424-SPO20pr (vector), or pRS424-SPO20pr-BoNT/En-LC-FLAG (BoNT/En-LC). Left panel: the cell lysates were subjected to Western blot analysis using anti-HA (Spo20/SNAP25(C)) and anti-actin antibodies. Right panel: the relative intensities of Spo20/SNAP25(C)-3HA detected in the cell lysates. The intensities of the Spo20/SNAP25(C)-3HA band were normalized to actin. The intensity of the Spo20/SNAP25(C)-3HA band detected in the cells harboring pRS424-SPO20pr was defined as 1. Data are presented as the mean ± SEM. Statistical significance was determined by two-tailed unpaired Student’s *t*-tests; *n* = 3 (**a**,**b**), *n* = 4 (**c**,**d**); * *p* < 0.05, *** *p* < 0.001, **** *p* < 0.0001, ns: not significant (*p* ≥ 0.05).

## Data Availability

The data supporting the findings of this study are available within the article and Appendix A. Further relevant data are available on request from the corresponding author.

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
