# Peer review of "Construction of a Yeast Cell-Based Assay System to Analyze SNAP25-Targeting Botulinum Neurotoxins"

_microorganisms, 2023, doi:10.3390/microorganisms11051125_

Round 1

Reviewer 1 Report

In this manuscript, the authors examine the yeast cell-based assay for analyzing of BoNT that targets SNAP-25. They found that the chimeric SNARE protein, consisting of Spo20 and SNAP25, is functional in the sporulation event of yeast cells. The chimera is sensitive to SNAP25-targeting BoNTs and their defects in the sporulation. An advance of this study is the claim that the activity of BoNTs can be assessed by their colorimetric assay based on yeast cells. Although this is an interesting study, there are several points that require more careful examination. The authors must address the following concerns.

Major comments

1.      A major shortcoming of this paper is that the authors fail to present appropriate controls. Actin and/or expression of BoNT-FLAG must be presented in all western blots. Following is an example. In Figure 2a, the authors should provide the actin as a loading control. In Fig 2d, the authors must present the amount of BoNT/C-FLAG and BoNT/CE230Q using an anti-FLAG antibody.

2.      Some of the western blots appear to be overexposed (or saturated), making it impossible to use them for quantitative analysis. The quantification does not reflect the actual band intensities (Figure 2d, 3b, 4c, and 5c). Details of how normalization was performed is lacking in the Methods section for sporulation assay.

3.      Details of how normalization was performed are lacking in the Methods section for sporulation assay.

4.      In the Discussion, the authors assert that their yeast-based assay can quantify the activity of BoNT. The assert should be toned down since this assay is the over-reliance on the overexpression of the BoNT. Precise normalization is necessary for the quantification of the activity of BoNT, because there are variations in the expression level of each LCs in yeast cells (In Figure 3a, 4b, 4d, and 5a).

Minor comments

1.      The authors should refer to serotypes of BoNT with appropriate citations in the “Introduction” section.

2.      It would be informative to see cleavage of Spo20/SNAP25 by BoNTs using anti-SNAP25 antibody. There is no direct data to show the chimeric SNARE is cleaved by BoNT.

3.      Line 146: Please indicate what amino acids from where to where in BoNT are cloned as the LC.

4.      The author must describe any approval for handling the BoNT-gene for the cloning work.

5.      Line 195: It would be helpful if the authors would explain the assay used to measure sporulation in more detail.

6.      Line 215: The authors should explicitly state what they mean by three independent “samples”. Are they denoting the culture on the same day, independent culturing day, etc?

7.      Line 243: The authors switch tags, GFP to HA tag, without any apparent rationale.

8.      Line 435: there is no report that the mutant BoNT/EnE226Q-LC is catalytically inactive.

9.      The labeling in all figures is confusing.

10.   “BoNT-FLAG” should be “LC-FLAG” in all Figures.

11.   It would be helpful if the authors would present the picture of the sporulation of yeast (vector, HA-Spo20, HA-Spo20/SNAP25(C)) in Figure 1b.

Correction

1.      Line 231; Correct “Figure S1f” to “Figure S1e”.

Author Response

I would like to thank all the reviewers for reviewing our manuscript and valuable comments. Response to reviewers are follows.

Major comments

  1. A major shortcoming of this paper is that the authors fail to present appropriate controls. Actin and/or expression of BoNT-FLAG must be presented in all western blots. Following is an example. In Figure 2a, the authors should provide the actin as a loading control. In Fig 2d, the authors must present the amount of BoNT/C-FLAG and BoNT/CE230Qusing an anti-FLAG antibody.

Response:

            Based on the reviewer’s comment, we have added loading control (actin) to all western blotting figures.

            Levels of all mutant BoNTs were compared to those of wild-type ones. The results have been shown in Figure S3. Expression levels of most BoNT mutants (except BoNT/A) were lower compared to those of wild-type ones. These results have been described in the test (line 304-305, 360-361, 405-406, 421-422, 472-473). Nevertheless, we think our results overall show that the BoNTs target appropriate chimeras in yeast cells. This description has been added in Discussion (line 530-534).

  1. Some of the western blots appear to be overexposed (or saturated), making it impossible to use them for quantitative analysis. The quantification does not reflect the actual band intensities (Figure 2d, 3b, 4c, and 5c). Details of how normalization was performed is lacking in the Methods section for sporulation assay.

Response:

            We have revised all western blotting analyses used to quantify expression levels. We have made sure that the images are not saturated. Description regarding the quantification has been revised based on the reviewer’s comment (line 225-228).

  1. Details of how normalization was performed are lacking in the Methods section for sporulation assay.

Response:

            The quantities of spores were correlated with the fluorescent levels of dityrosine (Figure S3a). Thus, measured vales of the microplate reader were directly used to quantify sporulation efficiency (2% potassium acetate solution was used as a blank control). This description has been added in the text (line 212-214).

  1. In the Discussion, the authors assert that their yeast-based assay can quantify the activity of BoNT. The assert should be toned down since this assay is the over-reliance on the overexpression of the BoNT. Precise normalization is necessary for the quantification of the activity of BoNT, because there are variations in the expression level of each LCs in yeast cells (In Figure 3a, 4b, 4d, and 5a).

Response:

            The reviewer’s suggestion is quite reasonable. Thus, we have revised Discussion. In the revised manuscript, we have described that ‘One advantage of this assay is that SNAP25-targeting BoNT-LCs are readily identified’ (line 505-506).

Minor comments

  1. The authors should refer to serotypes of BoNT with appropriate citations in the “Introduction” section.

Response:

            We have added the information in the text (line 28-29).

  1. 2. It would be informative to see cleavage of Spo20/SNAP25 by BoNTs using anti-SNAP25 antibody. There is no direct data to show the chimeric SNARE is cleaved by BoNT.

Response:

            It would be difficult to detect the digested fragment even if anti-SNAP25 antibodies were used. The result may further support our notion. But even without the result, we think that our results overall support the conclusion that the BoNTs target appropriate chimeras in yeast cells as.

  1. 3. Line 146: Please indicate what amino acids from where to where in BoNT are cloned as the LC.

Response:

            We have shown amino acid numbers in the Supplementary methods Plasmids and Table S1.

  1. 4. The author must describe any approval for handling the BoNT-gene for the cloning work.

Response:

            Based on the reviewer’s suggestion, Materials and Methods (Plasmid section) has been revised (line 92-93).

  1. 5. Line 195: It would be helpful if the authors would explain the assay used to measure sporulation in more detail.

Response:

            Based on the reviewer’s suggestion, Materials and Methods (Sporulation assay section) has been revised (line 204-208).

  1. 6. Line 215: The authors should explicitly state what they mean by three independent “samples”. Are they denoting the culture on the same day, independent culturing day, etc?

Response:

            Independent samples were prepared on different days and cells were obtained from different colonies. These have been included in the text (line 233-234).

  1. 7. Line 243: The authors switch tags, GFP to HA tag, without any apparent rationale.

Response:

            We have briefly described the reason in the text (line 261-262).

  1. 8. Line 435: there is no report that the mutant BoNT/EnE226Q-LC is catalytically inactive.

Response:

            We used the mutant as a putative inactive BoNT/En. Awe have revised the corresponding part of the (line 467-469).

9 and 10. The labeling in all figures is confusing / “BoNT-FLAG” should be “LC-FLAG” in all Figures.

Response:

            We have revised all figures (-FLAG has been removed; -LC has been added). We have also added LC after BoNT in the text if necessary.

  1. 11. It would be helpful if the authors would present the picture of the sporulation of yeast (vector, HA-Spo20, HA-Spo20/SNAP25(C)) in Figure 1b.

Response:

            We have shown the pictures in Figure S2.

Correction

  1. The Line 231; Correct “Figure S1f” to “Figure S1e”.

Response

            The mistake has been corrected.

Reviewer 2 Report

The manuscript is well designed and put together nicely. There are some typos here and there but explained well.

Author Response

We are grateful for your positive evaluation. We have revised typo mistakes.

Reviewer 3 Report

The aim of the manuscript entitled "Construction of a yeast cell-based assay system to analyze SNAP25-targeting botulinum neurotoxins" is to genetically modify Yeast into a cell-based assay to test botulinum neurotoxins' activities. While there was already in the field previously published work on genetically modifying yeast SNAREs proteins into chimere sensitive to BoNT (Sso1/syntaxin 1 and 78 Snc2/VAMP2 chimeras), this work focuses on producing Spo20/SNAP25 yeast functional chimere, which can be used for quantitatively assessing BoNTs activities through sporulation defect. The main novelty of this work is the possibility to quantify the activities of BoNTs through the digestion of the chimere Spo20/SNAP25.

The article is very well written, with clear English. The manuscript is clear, relevant to the field, and presented in a well-structured manner. All the different steps to produce a viable yeast expressing the chimere Spo20/SNAP25 are described in detail. I particularly appreciated that negative results were described and how they help improve the previous non-functional designs. 

I have very minor comments, see below.

1) In the methods section:

- The plasmids construction description is extremely long and difficult to read. I would suggest moving it as it is in the supplementary file and just summarizing it in the method section. You could for example keep the names of the plasmids and for what purpose they have been constructed.

2) In the Supplementary Materials section:

- this section is for referencing the supplementary files. I don't think it is necessary to have a detailed description of the content. Maybe you could summarize it the same way as for the plasmids in the methods section.

3) In the discussion:

This is a genetic engineering paper, and as it is the discussion focuses on the genetic aspect and hurdles of the design. I would have appreciated the author elaborating on the evolution implication of such chimere being functional, for example in the last paragraph.

Author Response

I would like to thank all the reviewers for reviewing our manuscript and giving us valuable comments. Response to reviewers are follows (line numbers are for the document with track changes).

  1. The plasmids construction description is extremely long and difficult to read. I would suggest moving it as it is in the supplementary file and just summarizing it in the method section. You could for example keep the names of the plasmids and for what purpose they have been constructed.

Response:

            Based on the reviewer’s suggestion, detailed methods to construct the plasmids have been described in Supplementary methods. Brief descriptions of the plasmids are also presented in Table S1.

  1. this section is for referencing the supplementary files. I don't think it is necessary to have a detailed description of the content. Maybe you could summarize it the same way as for the plasmids in the methods section.

Response:

            We have removed detailed descriptions of supplementary materials.

  1. This is a genetic engineering paper, and as it is the discussion focuses on the genetic aspect and hurdles of the design. I would have appreciated the author elaborating on the evolution implication of such chimere being functional, for example in the last paragraph.

Response:

            It is difficult to discuss about the evolution implication of the chimers in this paper. So, we have briefly described that the chimeric SNAREs maybe useful to gain insight into rthe egulation of synaptic SNARE proteins in Discussion (line 517-521).

Round 2

Reviewer 1 Report

The authors have responded appropriately to my concerns, providing appropriate controls and additional data.